# Mixed twitch and tetanus electrical stimulation via belt-electrode effectively attenuates denervation-induced muscle atrophy

Hiroyuki Uno[1]*, Mako Isemura[1], Shohei Kamiya[1], Ryuji Akimoto[1], Katsu Hosoki[1], Shunta Tadano[1], Karina Kouzaki[2], Yuki Tamura[2], Takaya Kotani[3], Koichi Nakazato[2]

1 HOMER ION Co., Ltd., Shibuya-ku, Tokyo, Japan, 2 School of Health and Sport Science, Nippon Sport Science University, Setagaya-ku, Tokyo, Japan, 3 Graduate School of Social Welfare, Tokyo University, Meguro-ku, Tokyo, Japan

* h.uno@homerion.co.jp

## Abstract

Belt electrode skeletal muscle stimulation (B-SES) is a method of applying electricity to contract muscles using belt-shaped electrodes. We previously reported that twitch contractions increase mitochondrial synthesis and suppress muscle proteolysis. In contrast, tetanus contraction increases muscle protein synthesis and suppresses muscle proteolysis. This study aimed to determine whether combining twitch- and tetanus-mode stimulations, which are known to differentially regulate mitochondrial and protein synthesis pathways, can more effectively attenuate muscle atrophy induced by denervation. Male Sprague-Dawley rats were subjected to acute or chronic B-SES. In the acute study, animals were assigned to control (CONT), tetanus (60 Hz), or Combined Stimulation (CS: 7–8 Hz for 15 min to 60 Hz for 3 min) groups. Four groups were tested in the chronic study: CONT, denervation (DEN), DEN + 60 Hz, and DEN + CS groups. Acute stimulation resulted in significantly lower muscle glycogen level, increased phosphorylated AMPK and p70S6K in the gastrocnemius muscle (GAS、n = 4) at 60 and CS compared to CONT, with no difference between 60 and CONT. After seven days, both muscle wet weight and cross-sectional area (CSA) were significantly reduced in the DEN group. Although both 60 Hz and CS attenuated atrophy, CS resulted in greater preservation (GAS CSA: DEN + CS, 71% CONT; DEN + 60, 61% CONT). In conclusion, the combination of different stimulation modalities (frequencies) was more effective than continuous tetanus stimulation in preventing denervation-induced muscle atrophy owing to an increase in muscle protein synthesis and inhibition of mitochondrial reduction.

## Introduction

Although electrical stimulation (EMS) of skeletal muscles is an effective treatment for muscle atrophy, detailed prescriptions such as frequency, duration, and current have

which permits unrestricted use, distribution, and reproduction in any medium, provided the original author and source are credited.

**Data availability statement:** All relevant data are within the manuscript and its Supporting Information files.

**Funding:** Research funding for this study was provided by HOMER ION Co., Ltd.

**Competing interests:** Authors with competing interests Research funding for this study was provided by HOMER ION Co., Ltd., H.U., K.S., R.A., K.H., S.T., and M.I. are HOMER ION Co., Ltd. employees, and K.N. is a co-researcher. The other authors have no financial disclosures related to this paper.

not been fully investigated [1–3]. Depending on the electrical frequency, twitch and tetanus contractions are induced by low and high electrical frequencies, respectively. The effects of the two contraction modes differed. Previous studies have suggested that twitch stimulation mimics aerobic exercise and enhances mitochondrial synthesis, whereas tetanus stimulation mimics resistance training and enhances muscle protein synthesis [4]. However, these two contraction modalities have not been effectively prescribed or distinguished to prevent muscle atrophy.

In animal models, tetanus electrical stimulation has been demonstrated to elicit resistance exercise-like adaptations, including the activation of protein synthesis signaling pathways and skeletal muscle hypertrophy [4–6]. In contrast, twitch-mode electrical muscle stimulation (EMS) has been shown to enhance mitochondrial mass and enzymatic activity, while concurrently suppressing muscle proteolytic signaling, which is typically upregulated during muscle atrophy [7,8].

Our previous studies revealed that both twitch and tetanus contraction stimulation via belt electrode skeletal muscle stimulation (B-SES) effectively attenuate muscle atrophy across multiple muscle groups. Specifically, twitch stimulation was found to promote mitochondrial biogenesis and increase mitochondrial content, while inhibiting proteolytic signaling. Conversely, tetanus stimulation enhanced muscle protein synthesis and suppressed proteolysis [9,10].

Moreover, tetanus stimulation has been implicated in the activation of the Akt/mTOR signaling pathway, which is associated with increased muscle mass and strength [11]. This pathway also plays a critical role in the transcriptional regulation of the ubiquitin ligases MuRF1 and Atrogin-1, which target muscle proteins for degradation via the 26S proteasome, thereby contributing to muscle atrophy [12]. Mitochondrial degradation is also considered a precursor to muscle atrophy and may contribute to the loss of muscle mass [13]. Muscle disuse alters metabolic function, leading to elevated production of reactive oxygen species (ROS), which subsequently activate catabolic pathways such as mitochondrial apoptosis and the ubiquitin-proteasome system, ultimately resulting in muscle atrophy [14]. These findings suggest that the regulation of muscle protein synthesis signaling may differ between tetanus and twitch stimulation, potentially involving distinct molecular pathways.

Given the distinct effects of twitch and tetanus contraction modes on atrophied muscle, their combination may elicit a different physiological response compared to continuous electrical stimulation using a single, fixed frequency.

Concurrent training has traditionally been modeled through combinations of voluntary exercise modalities, such as aerobic and resistance training, as well as hybrid approaches involving voluntary exercise and electrical stimulation—for example, tetanus stimulation applied during aerobic activity—which have been extensively investigated [15–17]. However, few studies have examined the combined application of twitch and tetanus stimulation in the context of involuntary exercise induced solely by electrical stimulation. Therefore, further research is warranted to elucidate the effects of this combined stimulation approach, particularly in populations such as elderly individuals, postoperative patients, and those with limited capacity for voluntary exercise.

In the present study, we employed a rodent B-SES model [9,10] to assess whether the combination of twitch and tetanus stimulation more effectively attenuates skeletal muscle atrophy compared to conventional tetanus stimulation alone. We also investigated the impact of this combined stimulation on mitochondrial biogenesis, muscle protein synthesis, and protein degradation.

Age-related skeletal muscle atrophy is closely associated with degeneration of the neuromuscular junction, which plays a pivotal role in the progression of atrophy [18]. Preventing atrophy in the context of denervation is considered essential for promoting recovery of skeletal muscle mass. Accordingly, we conducted these evaluations using a rat model of denervation-induced atrophy and hypothesized that combined twitch and tetanus stimulation would serve as an effective strategy to mitigate muscle atrophy by enhancing mitochondrial and muscle protein synthesis while suppressing protein degradation.

## Materials & methods

### Animals

Ten-week-old male Sprague-Dawley rats were purchased from CLEA (Japan). The mean body weight of all animals was 332.75 g ± 21.70 g (mean ± standard deviation (SD)). All rats were acclimatized by rearing in the environment described above for one week prior to the experiment. Acute and chronic belt electrode experiments were conducted in this study. All animals were randomly assigned as follows: acute stimulation experiment, chronic stimulation experiment n = 52; rats were kept in cages at 23°C with a 12 h/12 h light/dark cycle (dark time 18:00–06:00). All rats received a standard solid feed (CE-2; CLEA Japan, Tokyo, Japan) and water ad libitum.

### Electrical stimulation with belt electrodes

The right and left ankles were shaved under isoflurane anesthesia (anesthetic suction rate: 250−300 mL/min, concentration: ~2.0%−2.2%). The rats were placed on their backs on a table, and belt-type electrodes (Holmer Ion Corporation, Tokyo, Japan) were attached to both ankles (Fig 1a).

The electrical stimulation method used was modified from previous studies [9,10]. The muscles of the lower limbs were stimulated simultaneously on both sides with electrical stimulation for twitch (7–8 Hz) and tetanus (60 Hz). Stimulation intensity was preliminarily tested with a belt electrode and set to a minimum current run value of 3.0 mA (Fig 1b) (The current intensity of the twitch was set to 1.2 mA when the peak value of the waveform was aligned at 60 Hz), which produced maximum torque with 60 Hz stimulation. The lower limbs were not immobilized during contraction and were subjected to natural extension.

### Acute response to electrical stimulation with belt electrode

Acute stimulation and analysis were performed in three groups: control (CONT:n = 4), tetanus stimulation [60:n = 4], and concurrent electrical stimulation (CS:n = 4).

Under isoflurane anesthesia (anesthetic aspiration rate: 250–300 mL/min, concentration: ~2.0–2.2%) with reference to previously performed methods [9,10], acute stimulation with a belt electrode (60:60 Hz tetanus stimulation 5 min × 1 set, CS: 7–8 Hz twitch stimulation 15 min, followed by 1 min 30s shift to 60 Hz, 3 min tetanus stimulation. Fig 1c). Immediately following stimulation, the tibialis anterior (TA) and gastrocnemius (GAS) muscles were excised to assess glycogen content (GAS and TA) and phosphorylated AMPK levels (GAS) [19]. Six hours post-exercise, the TA and GAS muscles were harvested to evaluate phosphorylated p70S6K levels (n = 4) [20].

All experimental procedures were approved by the Animal Experimentation Committee of Japan Sport Sciences University (approval No. 017-A04). All procedures were conducted under anesthesia, and every effort was made to minimize animal suffering. The experiments were performed in accordance with the Fundamental Guidelines for Proper Conduct of Animal Experiment and Related Activities in Academic Research Institutions issued by the Ministry of Education, Culture, Sports, Science and Technology, Japan (No. 71, 2006).

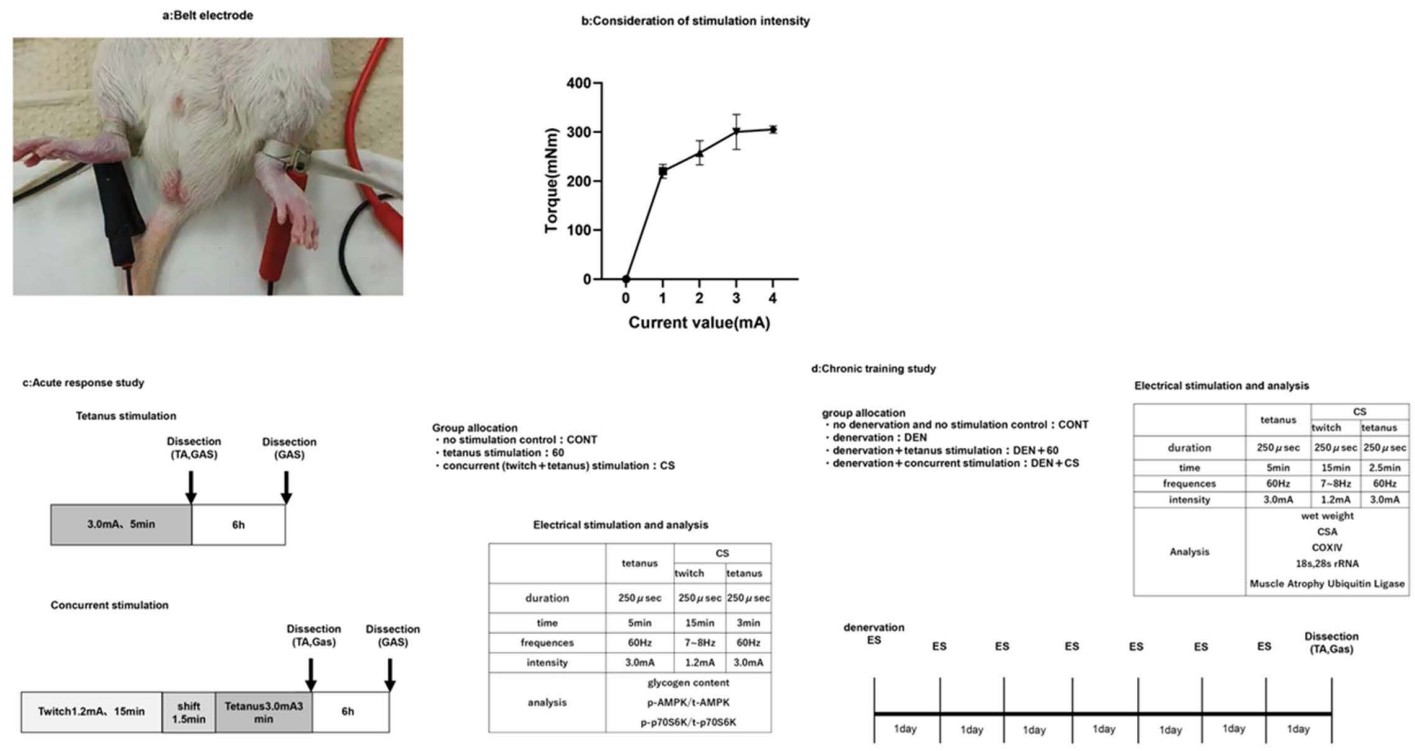

**Fig 1.  Belt electrode (a), preliminary experiment to determine the stimulus intensity (b) and experimental design (c,d).** (a) Belt electrodes were attached to anesthetized rats in the supine position. The left and right lower limbs were shaved, and the belt electrodes were attached to the saline-impregnated right and left ankles. (b) Preliminary experiment to determine the stimulus intensity. Since 3mA and 4mA were equivalent, 3mA was used as the stimulation intensity. (c) Acute and (d) chronic stimulation study design. Acute tibialis anterior (TA) and gastrocnemius (GAS) muscles were dissected immediately (glycogen content and phosphorylated AMPK assay) or 6 h after stimulation (phospho-p70S6K assay) to assess muscle glycogen content, phosphorylated p70S6K, and phosphorylated AMPK. Chronic stimulated TA and GAS muscles were dissected 24 hours after the last stimulation of the chronic response; TA and GAS were used for muscle weight and CSA analysis; GAS was used for Western blotting, electrophoresis, RT-PCR.

## Chronic response to electrical stimulation with belt electrodes

The chronic response to belt electrode stimulation was divided into four groups: control (CONT:n = 11), denervation (DEN:n = 11), denervation + 60 Hz electrical stimulation (DEN + 60:n = 6) and denervation + twitch followed by tetanus (DEN + CS:n = 6), with chronic response analysis. For electrical stimulation, chronic responses (60 Hz stimulation: 1 set of 5 min, 1 time/day for 7 days; combination stimulation: 1 set of 15 min twitch and 3 min tetanus, 1 time/day for 7 days) were performed under isoflurane anesthesia (anesthetic suction rate: 250−300 mL/min, concentration: ~2.0–2.2%), with the last TA and GAS harvested 24 h after stimulation (Fig 1d).

Muscle weight (n = 6–11) and muscle fiber cross-sectional area (CSA) (n = 6) were evaluated in the TA and GAS muscles. In addition, assessments of muscle atrophy inhibition, mitochondrial biogenesis, signal transduction, and ribosomal mass (n = 6) were conducted using the GAS muscle, which exhibited more inhibitory effect on muscle atrophy based on muscle weight measurements.

## Denervation

Surgical denervation was performed, as sciatic nerve denervation induces muscle atrophy. Under isoflurane anesthesia (aspiration rate: 250−300 mL/min, concentration: ~2.0–2.2%), the sciatic nerve was exposed through a small incision

on the posterior aspect of the right and left hind limbs. The sciatic nerve was transected by excising a few millimeters of the nerve using fine surgical scissors. The skin was closed using surgical glue. After denervation, a lack of mobility was observed in both legs in which the nerves were severed.

## Muscle

TA (muscle glycogen content, p70S6k, AMPK, muscle wet weight, cross sectional area) and GAS (muscle glycogen content, p70S6k, AMPK, muscle wet weight, cross sectional area, COXIV, rRNA, muscle atrophy ubiquitin ligase) were used. The muscles were cut into two pieces for biochemical analysis and CSA measurement of muscle fibers.

For the CSA measurements, the TA and GAS muscles were embedded in O.C.T compound (Sakura Finetech Japan, Tokyo, Japan) and frozen in cooled isopentane (166−00615, Fujifilm Wako Pure Chemicals Corporation, Osaka, Japan). Western blotting, PCR, and electrophoretic analysis were performed on GAS, which showed a greater effect of muscle atrophy inhibition in muscle weight; GAS was frozen in liquid nitrogen until analysis. All muscle samples were stored at −80°C until analysis.

## Muscle glycogen content measurement

Muscle glycogen content was measured as previously described [9,10]. Frozen muscle (10 mg-20 mg) was powdered and diluted in homogenization buffer containing 300 µl 30% KOH saturated with 100 µl 1M $Na_2SO_4$. The samples were boiled at 95°C and mixed every 10 min for 30 min.

After incubation, 480 µl of 99.5% ethanol was added and incubated at room temperature for 5 minutes. After incubation, samples were centrifuged at 5000 × g for 15 minutes at 4°C.

The supernatant was discarded, and the pellet was dried for 1 hour.

Next, 200 Tris-HCl (pH 6.8) was added, dissolved using a sonicator, and samples were incubated at 95°C for 2 hours. Finally, glycogen content was determined by measuring the absorbance at 505 nm using a Lab Assay TM Glucose Kit (298–65701, Fujifilm Wako Pure Chemicals Corporation, Osaka, Japan).

The data obtained were corrected by the weight of each muscle in powder form.

## Muscle fiber cross-sectional area

The CSA analysis was based on previous studies [21–23]. The medial GAS and TA muscles were cut into 10-µm-thick frozen sections using a cryostat (CM-502, Sakura Finetech Japan). Samples were blocked with Can Get Signal Blocking Reagent (NYPBR01, TOYOBO, Osaka, Japan) for 1 h and then incubated with anti-myosin heavy chain type I (1:500, BA-F8, DSHB, IA, USA), anti-myosin heavy chain type IIa (1:500, SC-71, DSHB), anti-laminin (L9393-1:500, Sigma Aldrich, MO, USA) at 4°C overnight. After incubation, the cells were washed with 0.1 M phosphate buffer (5 min × 3 times) and incubated with secondary antibodies (A3273-1:2000, A21147, A2112121-1:1000 A-11008, Invitrogen, California, USA) for 2 h. Images were captured using a confocal laser microscope (FV-3000; Olympus, Tokyo, Japan) and quantified using the MyoVision software (University of Kentucky, Tokyo, Japan).

## Protein extraction and Western Blotting

Western blotting was performed as previously described [9,10,24,25]. Muscle samples were homogenized in radioimmunoprecipitation assay (RIPA) buffer (188−02453, Fujifilm Corporation) containing a protease and phosphatase inhibitor cocktail (169–26063/167–24381, Fujifilm Wako Pure Chemicals Co. Protein concentrations of the samples were determined using the BCA method (297−73101, Fujifilm Wako Pure Chemicals Co. Equal amounts (40 µg) of protein were separated by SDS-PAGE (NW04127BOX, Thermo Fisher Scientific) and transferred to a polyvinylidene fluoride (PVDF) membrane (IB24001, Thermo Fisher Scientific). Protein transfer was confirmed using Ponceau S staining (33427.01;

SERVA Electrophoresis GmbH, Heidelberg, Germany). Membranes were blocked with a blocking reagent (NYPBR01, Toyobo, Osaka, Japan) for 1 h, and primary antibodies (9202,9205,2532,2535; Cell Signaling Technology, MA, USA and ab14744; Abcam, Cambridge, UK) were diluted with a dilution reagent (NKB-101, Toyobo). After incubation, the membranes were washed with Tris-buffered saline containing 0.01% Tween-20 (TBST; T9142; Takara Bio Inc. The membrane was then incubated with a secondary antibody (7074, Cell Signaling Technology) diluted with reagent (NKB-101, Toyobo) for 1 h at room temperature and washed again with TBST. The protein bands were visualized using a fluorescent reagent (SuperSignal West Pico chemiluminescent substrate; Thermo Fisher Scientific). iBright 1500 (FL1500, Thermo Fisher Scientific) and iBright Analysis Software (windows, Thermo Fisher Scientific) were used to scan and quantify the blots.

Ponceau S signal intensity was used as a loading control.

## Ribosomal RNA content

RNA was isolated from tissues as previously described [9,26,27]. Following this method, the muscle was homogenized with TRIzol reagent (Thermo Fisher Scientific) and separated into organic and aqueous phases using chloroform.

RNA was then separated from the aqueous phase using a kit (74106; Qiagen); RNA solution (5 μL) was mixed with 1 μL of GRR-1000GR Red Loading Buffer (Bio Craft, Tokyo, Japan) and electrophoresed on 1% of Tris-acetic ethylenediaminetetraacetic acid buffer. The 18S and 28S rRNA bands were scanned and quantified using a ChemiDoc XRS (170–9071, Bio-Rad) and Quantity One software (170–9600, version 4.5.2, Windows; Bio-Rad). Band intensities were corrected for the muscle sample volume.

## RT-PCR

RT-PCR was performed as previously described [9,10,28]. Muscle samples were homogenized using the TRIzol reagent (356203, Thermo Fisher Scientific). Chloroform (163–20145, Fujifilm Wako Pure Chemicals Corporation) was added to the homogenized samples, mixed, and allowed to stand for 15 min. Muscle samples were centrifuged at 4°C, 12,000 x g for 15 minutes. After collecting the supernatant, ethanol was added and mixed Total RNA was extracted using an RNA extraction kit (74106; QIAGEN, Hilden, Germany) Total RNA concentration was measured using NANODROP ONE (Thermo Fisher Scientific), and total RNA concentration was determined using an RNA extraction kit (74106), Reverse transcription was performed using cDNA. Real-time PCR was performed using the SYBR Gene Expression Assay (TOYOBO) in an optical reaction module equipped with a thermal cycler (CFX96, Bio-Rad, California, USA) and primers (Table 1).

βactin was used as a loading control.

## Statistical analysis

Data are presented as mean ± SD. Changes in glycogen content, phosphorylated AMPK, phosphorylated p70S6K, muscle wet weight, CSA measurements, mitochondria-related signals, muscle proteolytic signals, ribosomal synthesis signals, and rRNA content of acute and chronic stimuli were evaluated using parametric one-way ANOVA test Statistical significance was calculated as $p < 0.05$. Using Uncorrected Fisher's LSD correction for multiple comparisons [10]. Statistical significance was set at $p < 0.05$. GraphPad Prism (version 8.3.0, GraphPad Software, San Diego, CA, USA) was used for statistical analyses.

Table 1. Primer sequences used in RT-qPCR analysis.

| Target | Forward | Reverse |
|---|---|---|
| Atrogin-1 | AAGGAGCGCCATGGATACTG | AGCTCCAACAGCCTTACTACG |
| Murf1 | GACATCTTCCACGCTGCCAA | TGCCGGTCCATGATCACTTC |
| βactin | CACCCGCGAGTACAACCTTC | CCCATACCCACCATCACACC |

## Results

### Acute stimulation with B-SES reduced glycogen contents in skeletal muscles

We previously reported that tetanus or twitch B-SES induces lower extremity muscle contraction in muscle [9,10]. Therefore, we investigated whether concurrent stimulation activates the lower limb muscles and tetanus by measuring the glycogen content after tetanus and concurrent B-SES stimulation. Immediately after stimulation, glycogen levels were reduced in both the TA (anterior muscles) and GAS (posterior muscles) compared to controls at 60 and CS, with no significant difference between 60 and CS (Fig 2). These results suggest that B-SES results in glycogen consumption comparable to that of tetanus in the anterior and posterior muscles of both lower extremities. In addition, the two activation protocols recruited similar amounts of energy.

### Acute electrical stimulation induced phosphorylated p70S6K and phosphorylated AMPK

p70S6K phosphorylation improves muscle protein translation, leading to protein synthesis and muscle hypertrophy contribution [6,18]. Additionally, AMPK phosphorylation is induced by energy expenditure, which enhances mitochondrial biosynthesis [27]. We previously reported that tetanus contraction enhances the phosphorylation of p70S6K, and twitch contraction increases the phosphorylation of AMPK [9,10]. Therefore, we measured phosphorylated p70S6K and phosphorylated AMPK levels after tetanus and concurrent B-SES stimulation to examine whether they enhanced muscle protein and mitochondrial synthesis.

**Muscle glycogen content**

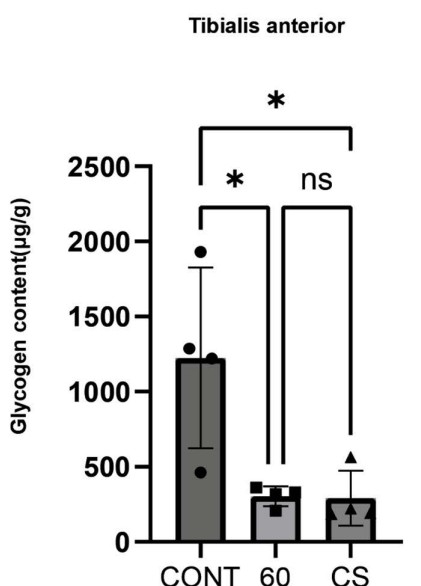
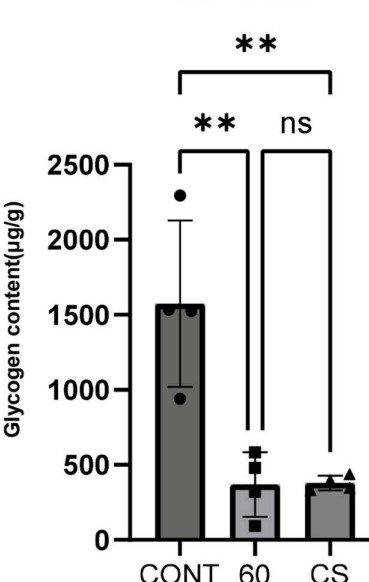

**Fig 2. Glycogen levels in TA and GAS after belt electrode acute stimulation.** Data are presented as mean ± SD. glycogen content of TA and GAS. Changes in the glycogen content of the lower limb muscles immediately after acute stimulation with the belt electrode were evaluated using a One-way ANOVA. CONT(n = 4): no stimulation control group; 60(n = 4):60 Hz electrical stimulation group with belt electrode; CS(n = 4): concurrent (twitch + tetanus stimulation) electrical stimulation group with belt electrode. *: $p < 0.05$, **: $p < 0.01$.

Immediately after stimulation, phosphorylated AMPK levels were elevated in both the TA and GAS muscles under the 60 Hz and CS conditions compared to the control, with no significant difference observed between the 60 Hz and CS groups. Six hours post-stimulation, phosphorylated p70S6K levels were increased in the TA and GAS muscles under both stimulation protocols compared to the control, again with no significant difference between the 60 Hz and CS groups (Fig 3). These findings suggest that both stimulation protocols activate signaling pathways involved in muscle protein synthesis and mitochondrial biogenesis.

## Chronic electrical stimulation attenuated denervation-induced muscle atrophy

During chronic stimulation, the effect of B-SES on sciatic nerve transection-induced muscle atrophy was first evaluated using muscle weight and muscle fiber cross-sectional area (CSA). The muscle weights and CSA of the atrophy by denervation (DEN), DEN+60, and DEN+CS TA and GAS groups were significantly lower than those of the control group, whereas the muscle weights and CSA of the DEN+60 and DEN+CS groups were significantly higher than those of the DEN group (Figs 4 and 5). (Body weight, CONT: 400.01±17.13 g, DEN: 363.30±16.83 g, DEN+60:371.83±10.56 g, DEN+CO: 375.04±18.48 g, mean±SD) Compared to DEN+60, both TA and GAS were significantly higher in DEN+CS (right TA showed significant trends), suggesting that the combination of twitch with following tetanus is effective in attenuating muscle atrophy.

## Mitochondrial content from chronic belt electrode skeletal muscle electrical stimulation

We have previously reported that twitch B-SES reduced the decrease in mitochondrial contents [9]. Therefore, in this study, we evaluated the mitochondrial content of the GAS muscle following chronic stimulation. COXIV protein, a marker of mitochondrial content, was significantly lower in the DEN, DEN+60, and DEN+CS groups than in controls. In contrast, the mitochondrial

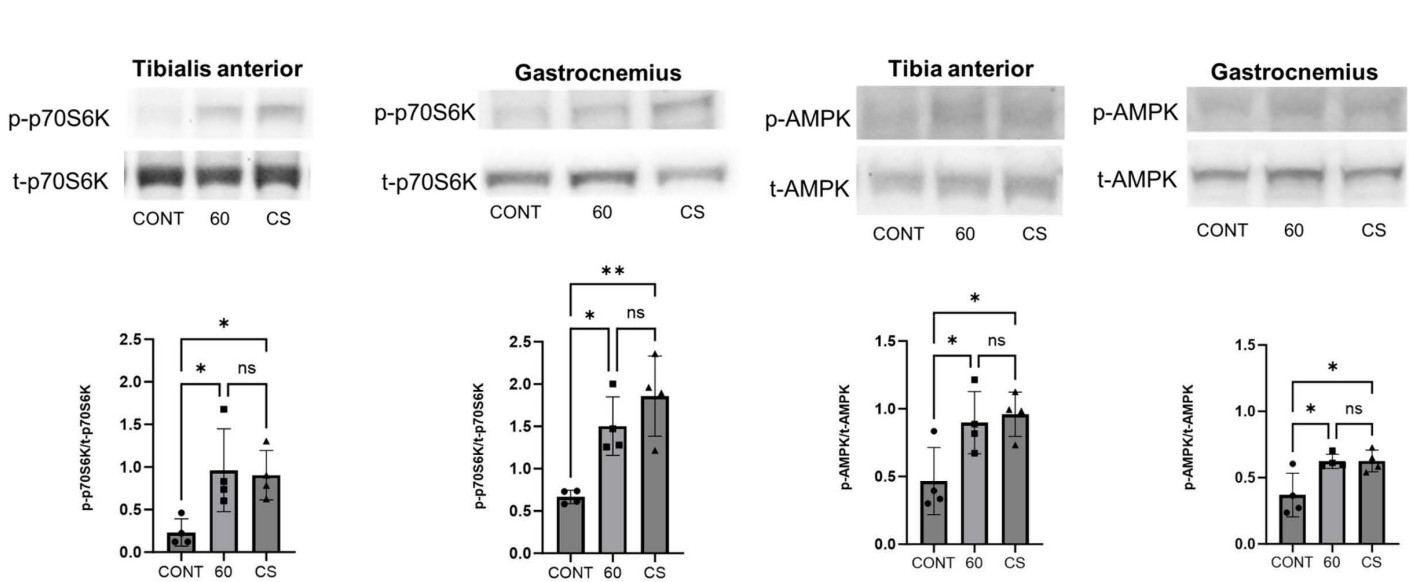

**Fig 3. Phosphorylated p70S6K (a) and phosphorylated AMPK (b) of TA and GAS after belt electrode acute stimulation.** Data are presented as mean±SD. Phosphorylated (a) p70S6K and (b) AMPK levels in the TA and GAS muscles. Phosphorylated p70S6K was assessed muscles of 6 hours after acute stimulation using a belt electrode, while phosphorylated AMPK was evaluated used musclesof immediately after stimulation. Statistical analysis was performed using one-way ANOVA. CONT (n=4): non-stimulated control group; 60 (n=4): 60 Hz electrical stimulation group using a belt electrode; CS (n=4): concurrent stimulation group (twitch+tetanus) using a belt electrode. *: p<0.05, **: p<0.01.

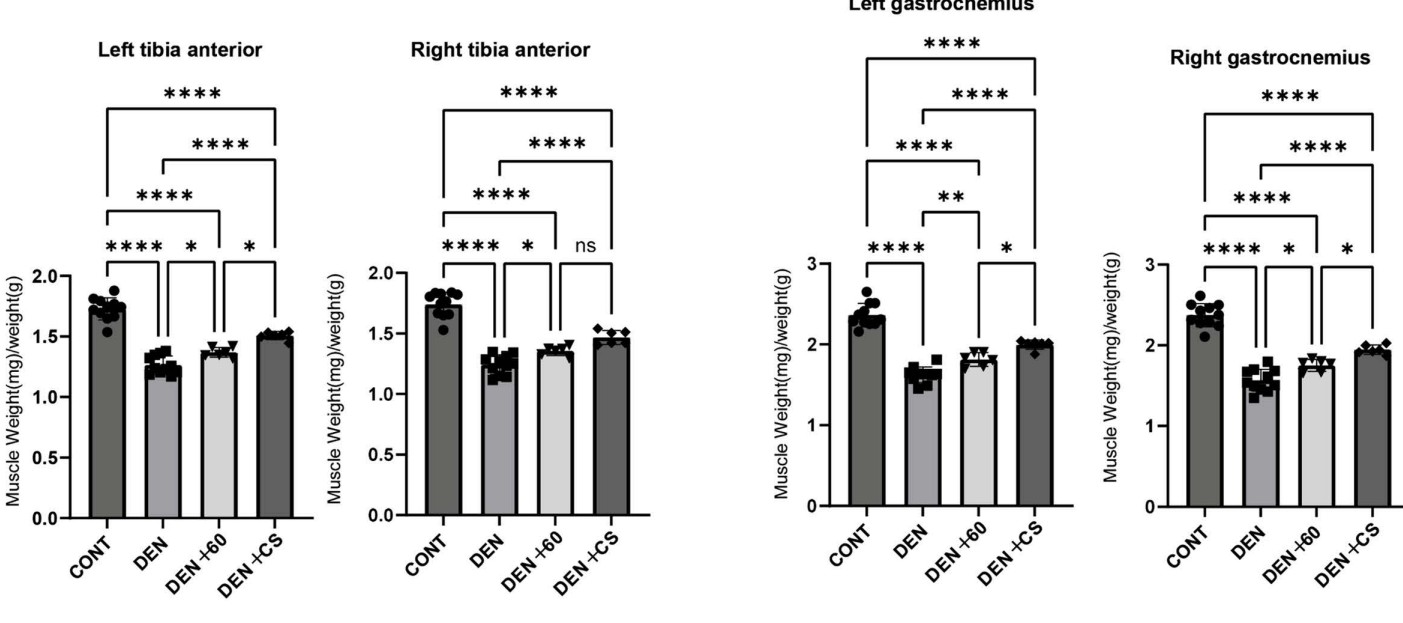

**Fig 4. Muscle wet weight of left and right TA and GAS after chronic stimulation with belt electrode.** Data are presented as mean ± SD. Changes in wet weight of lower limb muscles 24 h after chronic stimulation with the belt electrode were evaluated using a One-way ANOVA test. CONT(n = 11): no stimulation and no denervation control group; DEN(n = 11): denervation group; 60(n = 6): denervation + 60 Hz electrical stimulation group with belt electrode; CS(n = 6): denervation + concurrent (twitch + tetanus stimulation) electrical stimulation group with belt electrode. *: p < 0.05, **: p < 0.01, ***: p < 0.001, ****: p < 0.0001.

contents in the DEN + CS group were higher than those in the DEN and DEN + 60 groups, suggesting that the decrease in mitochondrial content was suppressed only by the combination of twitch and tetanus stimulation (Fig 6a). This suggests that the decrease in mitochondrial content upon denervation can only be suppressed by a combination of twitch and tetanus.

**Ribosomal content after chronic belt electrode skeletal muscle electrical stimulation**

The ribosomal RNAs 18S and 28S, which serve as markers of ribosome content in GAS muscle, were evaluated by electrophoresis following chronic stimulation. The 18S and 28S rRNA levels were significantly higher in the DEN, DEN + 60, and DEN + CS groups than in the CONT group and significantly higher in the DEN + 60 and DEN + CS groups than in the DEN group, with no significant difference between the DEN + 60 and DEN + C groups (Fig 6b). These results suggest that combined stimulation with a belt electrode enhances ribosome content and muscle protein synthesis.

**Muscle protein degradation Signals after chronic skeletal muscle electrical stimulation**

Muscle proteolytic signals of Atrogin-1 and MuRF1 were evaluated by RT-PCR after chronic stimulation.

Atrogin-1 expression was significantly higher in the DEN, DEN + 60, and DEN + CS groups compared to the CONT group. However, expression levels were significantly reduced in the DEN + 60 and DEN + CS groups relative to the DEN group. Additionally, a suppressive trend was observed in the DEN + CS group compared to the DEN + 60 group (p = 0.0801).

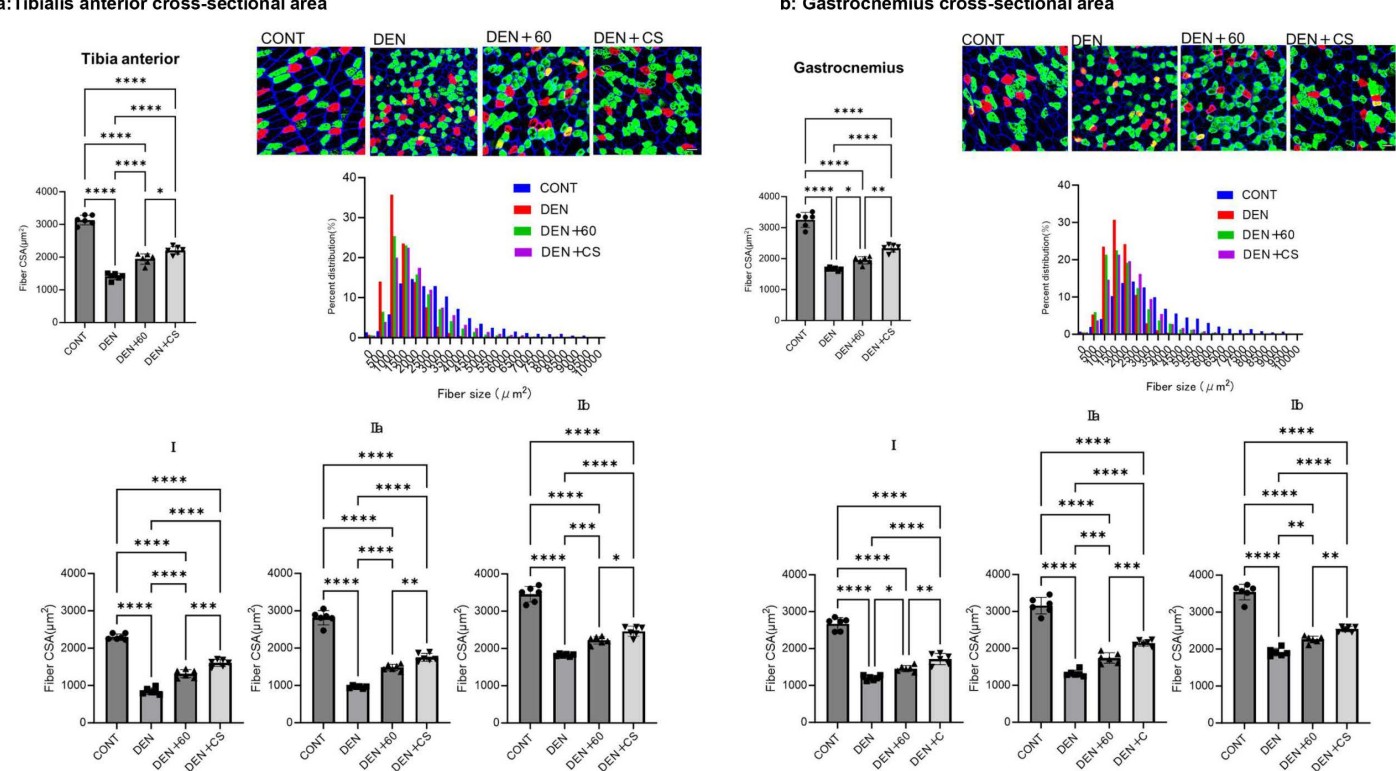

**a:Tibialis anterior cross-sectional area**

**b: Gastrocnemius cross-sectional area**

**Fig 5. Muscle fiber cross-sectional area (CSA) of left and right TA and GAS after chronic stimulation with belt electrode.** Data are presented as mean ± SD. Changes in the cross-sectional area (CSA) of TA and GAS muscle fibers in the lower limbs were evaluated 24 hours after chronic stimulation using belt electrodes. Statistical analysis was performed using one-way ANOVA. Blue staining indicates laminin, red indicates type I fibers, and green indicates type IIa fibers. Scale bar: 100 μm. CONT (n = 6): control group without stimulation; DEN (n = 6): denervation-only group; 60 (n = 6): denervation plus 60 Hz electrical stimulation using a belt electrode; CS (n = 6): denervation plus concurrent electrical stimulation (twitch + tetanus) using a belt electrode. *: p < 0.05, **: p < 0.01, ***: p < 0.001, ****: p < 0.0001.

MuRF1 expression was higher in the DEN, DEN + 60, and DEN + CS groups than in the CONT group; however, it was significantly attenuated in the DEN + 60 and DEN + CS groups compared to the DEN group, and there was no significant difference between the DEN + 60 and DEN + CS groups (Fig 7a and 7b). These results suggest that the increase in muscle proteolysis caused by denervation was suppressed by B-SES tetanus and combined stimulation. Importantly, concurrent stimulation tended to suppress proteolysis more effectively than tetanus stimulation alone.

## Discussion

In this study, we confirmed that the stimulation of rats with a combination of B-SES twitch and tetanus stimulation and fixed tetanus stimulation increased glucose consumption, p70S6K phosphorylation, and AMPK phosphorylation. Chronic stimulation for seven days attenuated denervation-induced muscle atrophy in several muscle groups. In particular, mixed-frequency tetanus stimulation is more effective than fixed-frequency tetanus stimulation. Furthermore, we confirmed that B-SES with mixed twitch and tetanus stimulation increased both ribosomal and mitochondrial contents. Furthermore, the increase in muscle protein degradation signaling induced by denervation was suppressed by both tetanus and concurrent (twitch + tetanus) stimulation using B-SES, with concurrent stimulation showing a greater suppressive tendency. Based on these results, we discuss the basic underlying molecular mechanisms.

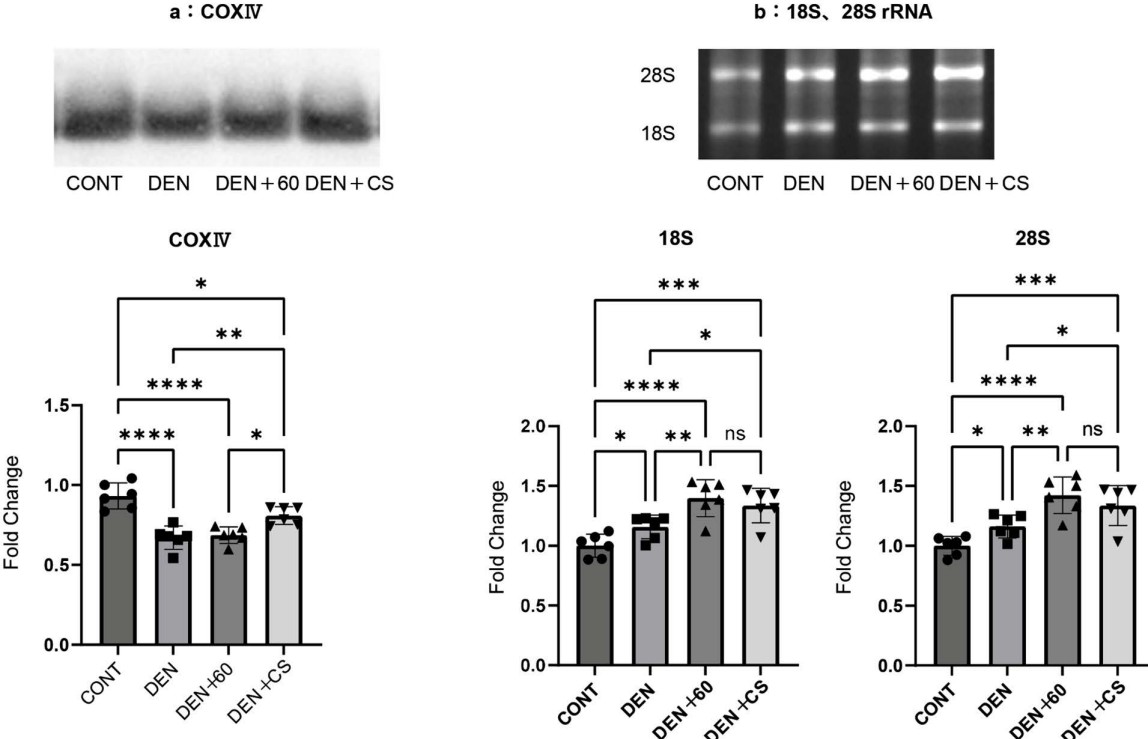

**Fig 6. CONXⅣ (a), 18S and 28S(b) after belt electrode chronic stimulation.** Data are presented as mean±SD. Changes in COX IV (a), and 18S and 28S rRNA levels (b) in the GAS muscle were evaluated 24 hours after chronic stimulation using belt electrodes. Statistical analysis was performed using one-way ANOVA. CONT (n=6): control group without stimulation; DEN (n=6): denervation-only group; 60 (n=6): denervation plus 60 Hz electrical stimulation using a belt electrode; CS (n=6): denervation plus concurrent electrical stimulation (twitch+tetanus) using a belt electrode. *: p<0.05, **: p<0.01, ***: p<0.001, ****: p<0.0001.

In this study, we first examined the dependence of the reduction in glycogen content, phosphorylated p70S6K, and phosphorylated AMPK on electrical stimulation.

Glycogen consumption was used as a measure of energy expenditure [29]. p70S6K phosphorylation is thought to lead to muscle protein synthesis and muscle hypertrophy [6,30], whereas AMPK phosphorylation is induced by energy expenditure during skeletal muscle contraction and promotes mitochondrial biogenesis [27,31]. As a result, all the stimulation protocols showed that muscle glycogen levels were significantly lower in 60 and CS than in CONT. The decreased levels in the two stimulation groups were similar. These results suggest that energy expenditure is similar in the tetanus and twitch-and-tetanus mixed protocols used in this study. Furthermore, levels of phosphorylated p70S6K and phosphorylated AMPK were also elevated, suggesting that both muscle protein synthesis and mitochondrial biogenesis may be similarly promoted. These findings indicate that acutely activated signaling pathways are not substantially affected by differences in electrical stimulation frequency. Increased phosphorylation of p70S6K and AMPK was observed in both the TA and GAS muscles, suggesting enhanced energy consumption and muscle protein synthesis in both muscle types. Our previous studies demonstrated that twitch-induced stimulation promotes mitochondrial biogenesis and that tetanus stimulation increases muscle protein synthesis [9,10]. Notably, AMPK, which plays a key role in mitochondrial biogenesis, was also elevated following tetanus stimulation. Since AMPK is known to facilitate glucose uptake [32], the observed increase in energy consumption may have contributed to this effect.

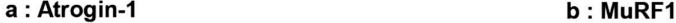

**Muscle Atrophy Ubiquitin Ligase**

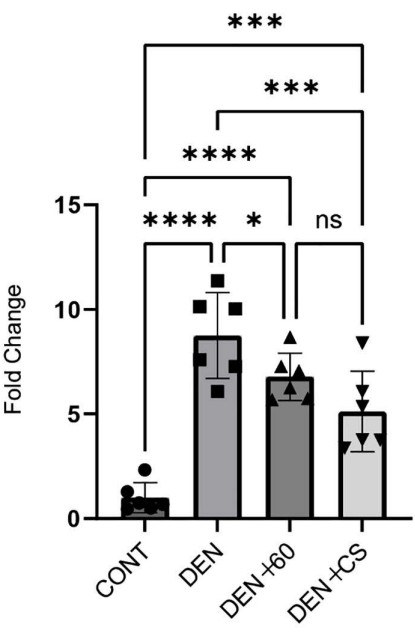

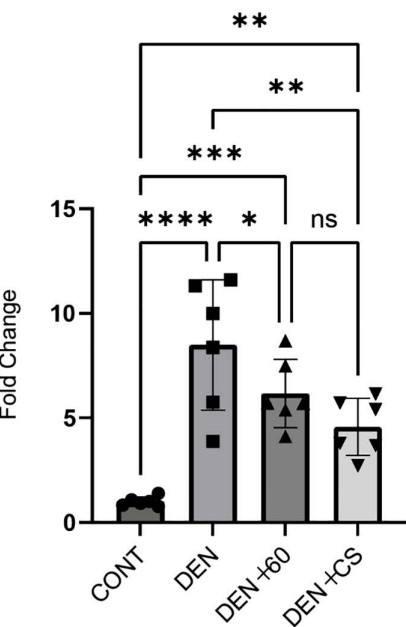

**DEN+60 vs DEN+CS=0.0801**

**Fig 7. Atrogin-1(a) and MuRF1(b) after chronic belt electrode stimulation.** Data are presented as mean±SD. Changes in Atrogin-1 (a) and MuRF1 (b) levels in the GAS muscle were evaluated 24 hours after chronic stimulation using belt electrodes. Statistical analysis was performed using one-way ANOVA. CONT (n=6): control group without stimulation; DEN (n=6): denervation-only group; 60 (n=6): denervation plus 60 Hz electrical stimulation using a belt electrode; CS (n=6): denervation plus concurrent electrical stimulation (twitch+tetanus) using a belt electrode. *: $p<0.05$, **: $p<0.01$, ***: $p<0.001$, ****: $p<0.0001$.

In a chronic study, denervated rats showed a decrease in muscle weight and CSA of the TA and GAS. In denervated rats, tetanus and concurrent stimulation showed significantly higher values for muscle weight and CSA, and significantly higher values for DEN+CS than for DEN+60, indicating that mixed-frequency stimulation had a greater effect on muscle atrophy.

In the left and right GAS muscles, significant inhibition of muscle atrophy was observed in both muscle weight and CSA compared to the 60 Hz electrical stimulation group. In the right tibialis anterior (TA), a trend toward inhibition in muscle weight was noted ($p=0.0928$), suggesting that the GAS muscle exhibited a stronger suppressive effect. Under acute stimulation conditions, the ratio of phosphorylated to total p70S6K (p/t p70S6K) in the CS group was significantly higher than in the CONT group, particularly in the GAS (TA: *, GAS:**). Since p70S6K is involved in muscle protein synthesis [6,17] and influences muscle size [6], these findings suggest that CS stimulation may contribute to the suppression of muscle weight loss in the GAS muscle.

Furthermore, it has been reported that the superficial layers of the TA and GAS muscles are primarily composed of type IIb fibers, while the deep layers consist mainly of type I and IIa fibers [33,34]. The results of this study indicate that atrophy was inhibited in both type I and type IIa/IIb fibers. Electrical stimulation induced hypertrophy regardless of muscle fiber

type [35], suggesting that it may influence fiber diameter in both superficial and deep muscle layers. In this study, both 60 Hz and concurrent stimulation (CS) protocols suppressed muscle atrophy in type I and type II fibers. Since 60 Hz stimulation does not induce single twitches, and both type I and type II fibers are known to increase in diameter with resistance training, it is likely that both twitch and tetanus stimulation contributed to hypertrophic effects across different muscle layers.

The mitochondrial content in skeletal muscles is reduced by inactivity [36,37]. It has been suggested that an accompanying increase in reactive oxygen species (ROS) promotes muscle protein degradation [38], thereby contributing to progressive muscle atrophy. COX IV, which showed that the content of mitochondria, was decreased by denervation as well as muscle weight, and no difference was seen in DEN + 60 compared to DEN. The decrease was not suppressed, but DEN + CS showed significantly higher values, indicating that the decrease was suppressed only by concurrent stimulation.

Skeletal muscle mass is also thought to be regulated by the balance between muscle protein synthesis and degradation [39–41]. Ribosomal content has been suggested to be related to muscle protein synthesis and muscle mass [42,43], with tetanus EMS activating ribosome synthesis and increasing ribosomal content [26]. We previously reported that twitch with B-SES inhibited the decrease in mitochondrial content and the increase in muscle protein degradation, and that tetanus inhibited the increase in ribosome content and muscle protein degradation [10]. Therefore, we evaluated the mitochondrial content, ribosomal content, and muscle proteolytic signals after 1 week of chronic stimulation. In this study, the 18S and 28S ribosomal contents were significantly increased in the DEN + 60 and DEN + CS groups compared to those in the CONT and DEN groups for both 18S and 28S. Regarding protein metabolism, the expression of the muscle proteolytic signals Atrogin-1 and MuRF1 was upregulated by denervation, but the upregulation of Atrogin-1 was suppressed in the DEN + 60 and DEN + CS groups. Furthermore, concurrent stimulation tended to suppress proteolysis more effectively than tetanus stimulation alone. It has been suggested that tetanus stimulation activates the Akt/mTOR pathway, which may contribute to increased muscle mass and strength [11]. The Akt/mTOR pathway also plays a role in the transcriptional regulation of the ubiquitin ligases MuRF1 and Atrogin-1, which mediate the ubiquitination and subsequent degradation of muscle proteins via the 26S proteasome, leading to muscle atrophy [12]. Additionally, mitochondrial dysfunction has been proposed as a precursor to muscle atrophy, contributing to the reduction in muscle mass [13]. Muscle disuse is known to alter metabolic function and increase the production of reactive oxygen species (ROS), which in turn activates catabolic pathways such as mitochondrial-mediated apoptosis and the ubiquitin–proteasome system, ultimately resulting in muscle atrophy [14].

Our previous research demonstrated that denervation reduces mitochondrial mass and activity, as indicated by decreased cytochrome c levels and citrate synthase activity. Therefore, concurrent stimulation with both twitch and tetanus stimulation appears to inhibit the expression of ubiquitin ligases through an additional pathway, beyond the protein degradation-suppressing mechanism activated by tetanus stimulation alone, thereby enhancing the overall inhibitory effect.

The results of this study demonstrated that combined twitch and tetanus electrical stimulation using a belt electrode was more effective in suppressing denervation-induced muscle atrophy than tetanus stimulation alone. This enhanced effect may be attributed to the simultaneous activation of muscle protein synthesis via tetanus stimulation and the inhibition of mitochondrial content loss, which was not observed with tetanus stimulation alone. The preservation of mitochondrial content may have contributed to a greater suppression of muscle proteolytic signaling.

However, resistance exercise has been reported to enhance the molecular signaling of mitochondrial biogenesis induced by aerobic exercise and the effects of aerobic exercise on muscle fiber diameter and mTOR protein expression during strength training [44,45]. However, concurrent training may attenuate the maximal muscle strength and muscle hypertrophy produced by resistance training [46]. This interference leads to increased protein synthesis and muscle hypertrophy with resistance exercise [6,30,47] activation of the mechanical targets of rapamycin complex 1 (mTORC1) signaling is suppressed by the aerobic exercise-induced enhancement of mitochondrial biogenesis-induced adenosine 1-phosphate kinase [48]. Aerobic exercise after resistance exercise suppresses mTORC1 signaling, and interference

effects have been reported [17]. It has been suggested that the attenuation of mTORC1 activation by resistance exercise during concurrent training may be mediated by the AMPK signaling pathway. Moreover, the sequence of exercise modalities may be an important factor influencing the effects of concurrent training on muscle hypertrophy [49].

AMPK phosphorylation is observed during and immediately after exercise, but rapidly returns to baseline levels thereafter [24]. In contrast, phosphorylation of p70S6K (a downstream target of mTOR) gradually increases and reaches a significant level up to 6 hours after the cessation of exercise [25]. These findings suggest that the absence of interference in this study may be attributed to the application of twitch stimulation prior to tetanus stimulation. When tetanus stimulation precedes twitch stimulation, AMPK phosphorylation may occur during the elevation of mTORC1 activity, potentially attenuating mTORC1 signaling. In contrast, twitch stimulation followed by tetanus stimulation may allow p70S6K phosphorylation to occur after AMPK activity has returned to baseline, thereby avoiding interference.

From a molecular perspective, twitch stimulation mimics aerobic exercise, while tetanus stimulation resembles resistance exercise. Thus, the sequence of electrical stimulation may be a critical factor in determining the effects of concurrent stimulation on muscle hypertrophy. The order of twitch and tetanus stimulation appears to influence the physiological response, and may play an important role in modulating muscle adaptation. These differences may be influenced by multiple factors, including exercise sequence, rest intervals, and stimulation intensity.

In this study, no interference effect was observed with the simultaneous application of twitch and tetanus electrical stimulation. There were no significant differences in p70S6K phosphorylation levels or ribosome content compared to tetanus stimulation alone, while a trend toward greater suppression of muscle protein degradation signaling was noted. These findings suggest that the concurrent use of twitch and tetanus stimulation may offer a more effective rehabilitation strategy for preventing muscle atrophy, particularly in postoperative or elderly patients for whom voluntary training and long-term adherence are challenging.

As mentioned earlier, there are two primary types of electrical stimulation: twitch and tetanus. Tetanus is considered to mimic resistance exercise, while twitch resembles aerobic exercise, and each is thought to exert distinct effects on muscle physiology.

Therefore, in clinical applications, it is important to select stimulation modes based on the desired physiological outcomes—whether mimicking resistance or aerobic exercise. In this study, continuous concurrent stimulation was found to both inhibit the decline in energy metabolism and enhance muscle protein synthesis, suggesting that aerobic- and resistance-like effects can be achieved simultaneously within a single stimulation mode. This may be particularly beneficial in cases where voluntary training or long-term adherence is difficult, such as in postoperative or elderly patients.

From a clinical perspective, tetanus stimulation is relatively intense, and individuals who are not accustomed to electrical stimulation may find it uncomfortable or may require time to adapt. However, in this study, twitch stimulation was applied prior to tetanus stimulation, which is thought to be more tolerable. This approach may help subjects acclimate to the stimulation, allowing for a gradual increase in intensity and potentially enhancing the overall effectiveness of the intervention.

However, it is possible that some protocols may have greater suppress muscle atrophy effects or weaker effects owing to interference or enhanced effects. Future studies may provide electrical stimulation with greater ability to suppress muscle atrophy.

Furthermore, this study employed a seven-day protocol initiated immediately following denervation, targeting the early phase of muscle atrophy. While this approach may reflect the impact of early-stage intervention, the effects of prolonged intervention or variations in its timing remain unclear. Future investigations into the protocol design, timing, and duration of intervention may facilitate the development of more effective strategies and expand potential clinical applications.

## Supporting information

**S1 Raw Image.  Western blot and electrophoresis images.**
(PDF)

## Acknowledgments

We would like to thank Editage (www.editage.com) for English language editing.

## Author contributions

**Conceptualization:** Hiroyuki Uno.

**Data curation:** Hiroyuki Uno.

**Formal analysis:** Hiroyuki Uno.

**Investigation:** Hiroyuki Uno, Mako Isemura.

**Project administration:** Hiroyuki Uno.

**Resources:** Shohei Kamiya, Ryuji Akimoto, Katsu Hosoki, Shunta Tadano.

**Supervision:** Shohei Kamiya, Ryuji Akimoto, Koichi Nakazato.

**Visualization:** Hiroyuki Uno.

**Writing – original draft:** Hiroyuki Uno.

**Writing – review & editing:** Karina Kouzaki, Yuki Tamura, Takaya Kotani, Koichi Nakazato.

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
