## [Decision Letter · Decision Letter 0]

14 Jul 2025

PONE-D-25-28377Mixed twitch and tetanus electrical stimulation via belt-electrode effectively attenuates denervation-induced muscle atrophy in multiple muscle groups.PLOS ONE

Dear Dr. Uno,

Thank you for submitting your manuscript to PLOS ONE. After careful consideration, we feel that it has merit but does not fully meet PLOS ONE’s publication criteria as it currently stands. Therefore, we invite you to submit a revised version of the manuscript that addresses the points raised during the review process.

We look forward to receiving your revised manuscript.

Kind regards,

Atsushi Asakura, Ph.D

Academic Editor

PLOS ONE

Journal Requirements: 

4. To comply with PLOS ONE submissions requirements, in your Methods section, please provide additional information regarding the experiments involving animals and ensure you have included details on (1) methods of sacrifice, (2) methods of anesthesia and/or analgesia, and (3) efforts to alleviate suffering.

6. Thank you for stating the following financial disclosure:

 [Research funding for this study was provided by HOMER ION Co., Ltd.,]. 

7.  Thank you for stating the following in the Competing Interests section:

[Authors with competing interests

Research funding for this study was provided by HOMER ION Co., Ltd., H.U., K.S., R.A., K.H., S.T., and M.I. are HOMER ION Co., Ltd. employees, and K.N. is a co-researcher. The other authors have no financial disclosures related to this paper.].   

We note that one or more of the authors are employed by a commercial company: name of commercial company.

Within your Competing Interests Statement, please confirm that this commercial affiliation does not alter your adherence to all PLOS ONE policies on sharing data and materials by including the following statement: ""This does not alter our adherence to  PLOS ONE policies on sharing data and materials.” (as detailed online in our guide for authors http://journals.plos.org/plosone/s/competing-interests) . If this adherence statement is not accurate and  there are restrictions on sharing of data and/or materials, please state these. Please note that we cannot proceed with consideration of your article until this information has been declared.

8. Please include your full ethics statement in the ‘Methods’ section of your manuscript file. In your statement, please include the full name of the IRB or ethics committee who approved or waived your study, as well as whether or not you obtained informed written or verbal consent. If consent was waived for your study, please include this information in your statement as well.

Reviewers' comments:

Reviewer's Responses to Questions

**Comments to the Author**

1. Is the manuscript technically sound, and do the data support the conclusions?

Reviewer #1: Yes

Reviewer #2: Yes

Reviewer #3: Partly

2. Has the statistical analysis been performed appropriately and rigorously? 

Reviewer #1: Yes

Reviewer #2: Yes

Reviewer #3: Yes

3. Have the authors made all data underlying the findings in their manuscript fully available?

Reviewer #1: No

Reviewer #2: Yes

Reviewer #3: No

4. Is the manuscript presented in an intelligible fashion and written in standard English?

Reviewer #1: Yes

Reviewer #2: Yes

Reviewer #3: Yes

5. Review Comments to the Author

Reviewer #1: The authors showed that a combination of twitch and tetanus contractions was effective in suppressing muscle atrophy induced by sciatic nerve denervation. These findings may contribute to the development of new intervention strategies for muscle atrophy. Various analyses were carefully conducted, and the framework of the study is well understood. However, I have several concerns as follows.

1. The introduction is redundant; it is unclear in L55-81 whether this study is an extension of your research or whether you are trying to establish a new experiment based on previous research. The aim of the study should be clearly indicated.

2. A variety of factors can cause muscle atrophy, but the authors used the peripheral neuropathy model in this study. However, there is no research introduction on the use of this model. Please add this point to the Introduction section.

3. The Methods and Materials section describes the animals as follows: "Acute stimulation experiment, chronic stimulation experiment, n=52." Could you please provide the specific number of animals in each group? Similarly, when stating that acute stimulation experiments were performed in three groups and chronic stimulation experiments were divided into four groups, the number of rats in each group should be specified.

4. For Western blotting, is a loading control or total protein amount quantified at the same time to eliminate differences in protein expression between samples and the effects of manipulation during the experiment? The authors should add these to the Material & Methods section.

5. The authors described “Stimulation intensity was preliminarily tested with a belt electrode and set to a minimum current run value of 3.0 mA (1.2 mA for twitch), which produced maximum torque with 60 Hz stimulation.” The authors should add the results of the preliminary investigation to Material & Methods section (or supplemental files).

6. In “Acute response to electrical stimulation with belt electrodes” section, why did the authors examine the glycogen content and phosphorylated AMPK immediately after exercise and phosphorylated p70S6K in 6 h after exercise? Please provide a reason for the time to harvest the skeletal muscles. The authors did not assess the phosphorylated AMPK and p70S6K in TA. This investigation was titled “multiple muscle groups”, the lack of these results is a primary concern for this study.

7. After denervation, the sciatic nerve may sometimes reattach. To avoid this, the proximal and distal portions are often removed a few millimeters from the amputated area. Did the authors perform this procedure? Or, did they confirm that no reattachment occurred by dissection?

8. Since the total RNA concentration differs from sample to sample, the amount of RNA contained in the RNA solution (5 μL) differs for each sample. Fortunately, the authors have determined the total RNA concentration by Nanodrop; please provide the total RNA concentration of each sample used for the 18S and 28S searches and show that there is no difference between groups.

9. Regarding RT-PCR, how did the authors quantify Atrogin-1 and MuRF1 mRNA expression? Did you use the internal control? If so, the primer sequences should be listed in Table 1.

10. It is difficult to confirm inter-individual variation in the mean±standard error. Please indicate the data as mean±standard deviation (SD). The authors should also add about the post-hoc tests. Furthermore, much data was listed as folds. Please re-describe all the data in mean±SD.

11. In Figure 3, the band images of p-p70S6K and p-AMPK are too weak to confirm; please re-prepare the band images along with t-p70S6K and t-AMPK. Furthermore, two bands can be identified in p-p70S6K. Which of the two bands did you adopt?

12. In Figure 4, why did you separate the muscle wet weight into right and left sides? I suppose they should be combined. In addition to muscle wet weight, please add data on body weight and relative weight ratio.

13. In Figure 5, which color indicates which muscle fiber type? Furthermore, since the scale bar is not listed, the sizes cannot be compared. Why is this data listed as a percentage while the muscle wet weight is listed as raw data? All data should be presented as raw data, not as ratios.

14. Which muscles did the authors use for analyses of COXIV, 18S, 28S, Atrogin-1, and MuRF1? You must indicate all the data of the above parameters in TA and GAS.

15. There is confusion between the terms “Murf1” and “MuRF1” (L355, 361). Please unify the descriptions.

16. In Figure 6, did the authors use the loading control (e.g., GAPDH)

17. The contents of L385-392 are duplicated in the Introduction section and are not necessary.

18. The authors described “muscle atrophy progresses from extracellular matrix degradation due to increased MMPs associated with decreased mitochondrial content.” (L410-412) I can’t understand this cascade. The author states that the degradation of muscle component proteins is a cause of muscle atrophy, and in this context, the degradation of the extracellular matrix (ECM) is a key factor. The primary function of MMPs is to degrade the ECM, and they are not directly involved in the degradation of muscle component proteins. This point should be described in more detail, and previous studies should be cited.

19. The authors described “twitch EMS increases mitochondrial mass and enzyme activity while suppressing muscle proteolytic signaling” in the Introduction section (L57-58). However, even though COXIV was significantly higher in DEN+CS than in DEN+60, there was no significant difference in 18S and 28S between the two groups. Furthermore, surprisingly, there was no significant difference in Atrogin-1 and MuRF1 between these two groups, despite significant differences in CSA for each muscle fiber type in GAS and TA. The author must add your idea on this discrepancy in the Discussion section.

20. Considering the title of this study, the authors likely aim to demonstrate the effectiveness of twitch and tetanus contractions on multiple skeletal muscles. This is understandable given that the study selected the TA (flexor) and GAS (extensor), which have distinct functional roles, as sample muscles. Additionally, since both skeletal muscles exhibit significant differences in muscle fiber size between their deep and superficial layers, the authors may have also wanted to investigate this point.

I read the discussion section expecting these points to be addressed, but unfortunately, there was not much discussion on the above topics. I would like you to reconsider the results and discussion regarding the following points.

1. Discuss whether any characteristic differences were found in the results of the tibialis anterior and gastrocnemius muscles between the acute and chronic experiments. Regardless of whether differences were found or not, discuss your findings in relation to previous studies.

2. Please discuss the differences in effects between the superficial and deep layers of the tibialis anterior and gastrocnemius muscles (if necessary, present the results separately for the superficial and deep layers). Based on the content of above 1, please discuss the effects of twitch and tetanus in the superficial and deep layers.

Reviewer #2: 1. This is a well-designed study combining twitch and tetanus stimulation to combat muscle atrophy. The idea is novel and valuable, especially for clinical applications in populations with limited mobility. However, please more clearly differentiate how your combined approach surpasses traditional methods using only tetanic stimulation, especially in terms of clinical scalability or mechanistic superiority.

2. The acute and chronic study arms are well constructed. Still, it would help to include a schematic diagram summarizing group allocation, stimulation protocols (e.g., duration, frequency, intensity), and outcome measurements.

3. Auhors shown a good evidence that mixed stimulation enhances both mitochondrial and protein synthesis markers. Still, consider adding a discussion on possible molecular pathways that explain the additive/synergistic effects.

4. While results are promising, some comparisons (e.g., between DEN+60 and DEN+CS) show only trends, not significant differences. Please clarify which results are statistically significant and which are not.

5. The figures are informative, but some (especially Western blot and CSA images) appear low-resolution. Please upload clearer images and ensure consistent formatting of figure legends.

6. Include exact p-values wherever possible and define the number of animals (n) per group for each analysis in both the text and figure legends.

7. The manuscript reads well overall, but a few sections (especially the Results and Discussion) would benefit from language polishing for clarity and flow.

8. Author mention future use in postoperative or elderly patients. It would be great to briefly explain how this method could be implemented practically in clinical or rehabilitation settings.

9. Please discuss limitations such as the short duration (7 days) of chronic stimulation and whether longer-term studies or human trials are being considered.

10. Maintain consistent terminology (e.g., use "TA" and "GAS" uniformly, and define all abbreviations on first use) to avoid confusion for non-specialist readers.

Reviewer #3: This study presents interesting evidence suggesting that the combination of twitch and tetanus stimulation may be more effective in suppressing muscle atrophy. The findings are valuable and provide a meaningful contribution to the field of muscle stimulation research.

Introduction

Describe in detail the molecular mechanisms by which combined stimulation therapy was predicted to more effectively inhibit denervation sarcopenia compared to twitch or tetanus stimulation alone at the planning stage of the study.

Methods

Please provide more detailed information regarding the rationale for selecting the stimulation parameters used in this study, including the frequency and temporal distribution of the combined stimuli.

Results

Muscle wet weight may have been affected by each rat’s variability, such as body weight. Please provide the muscle wet weight and body weight data for each rat at the time of sacrifice.

Histological Evaluation

In the CSA graphs, the black dots appear to represent ratios; however, the definition of the comparison target is unclear. Do these values represent, within each muscle, the total cross-sectional area of each muscle fiber type, or the average cross-sectional area per fiber type?

To better visualize fiber size changes, I recommend adding muscle fiber CSA distribution plots, as shown in your previous paper.

The composition of muscle fiber types (fast-twitch and slow-twitch) has been reported to change in sarcopenia. And electrical stimulation affects the composition of muscle fiber type (Shi H, J Neuroeng Rehabil, 2023). Please provide supplemental data on fiber type composition (e.g., number and proportion of each fiber type) in each muscle examined.

Western Blotting

In the supplementary data, it is unclear which bands correspond to the same samples between p-p70S6K and t-p70S6K, as well as between p-AMPK and t-AMPK. The correspondence between these bands should be clearly indicated. Additionally, loading controls such as GAPDH or β-actin are not shown, making it difficult to confirm equal protein loading. Please include these controls.

Discussion

Although the description of the effects and mechanisms of B-SES is informative, some of this content may be more appropriately presented in the Introduction section.

The absence of interference between twitch and tetanus stimulation is a significant finding. Please expand the discussion on the molecular mechanisms that may underlie this result.

The combination of twitch and tetanus stimulation more suppressed the expression of muscle degradation markers. It would be informative if the authors could discuss the possible molecular pathways involved in this mechanism.

6. PLOS authors have the option to publish the peer review history of their article (what does this mean? ). If published, this will include your full peer review and any attached files.

**Do you want your identity to be public for this peer review?** For information about this choice, including consent withdrawal, please see our Privacy Policy .

Reviewer #1: No

Reviewer #2: **Yes: ** Dr Dinesh Kumar

Reviewer #3: No

---

## [Author Response · Author response to Decision Letter 1]

26 Aug 2025

Editor

Comment 1

Reply

Thank you for your check and reply. The PLOS ONE style requirements (including file naming requirements) were reviewed, and the manuscript was revised.

Comment 2

Reply

Thank you for the comment. During the antibody reaction, the membrane was cleaved at the desired molecular weight. Therefore, the membrane is very small.

The original, uncropped and unadjusted images are provided in supplemental information. The above explanation is provided in supplemental information.

Comment 3

Reply

Thank you for the comment. I accessed the ORCID site, created a new ID, and confirmed it in the Editorial Manager.

Comment 4

4. To comply with PLOS ONE submissions requirements, in your Methods section, please provide additional information regarding the experiments involving animals and ensure you have included details on (1) methods of sacrifice, (2) methods of anesthesia and/or analgesia, and (3) efforts to alleviate suffering.

Reply

Thank you for the comment. For experiments involving animals, (1) the method of sacrifice, (2) the method of anesthesia and/or analgesia, and (3) efforts to alleviate suffering were added to the Animal section of Materials & Methods.

Comment 5

Reply

Thank you for the comment. Funding for this research was provided by HOMER ION Laboratory Co., Ltd. as joint research funding, not an award.

We have corrected inconsistencies in the grant information provided in the "Funding Information" and "Financial Disclosures" sections.

Comment 6

6. Thank you for stating the following financial disclosure:

[Research funding for this study was provided by HOMER ION Co., Ltd.,].

Reply

Thank you for the comment. Research funding for this study was provided by HOMER ION Laboratory Co., Ltd. as a collaborative research grant.

The roles of authors who are employees of HOMER ION Laboratory Co., Ltd. in this study have been described in the cover letter.

Comment 7

7. Thank you for stating the following in the Competing Interests section:

[Authors with competing interests

Research funding for this study was provided by HOMER ION Co., Ltd., H.U., K.S., R.A., K.H., S.T., and M.I. are HOMER ION Co., Ltd. employees, and K.N. is a co-researcher. The other authors have no financial disclosures related to this paper.].

We note that one or more of the authors are employed by a commercial company: name of commercial company.

Within your Competing Interests Statement, please confirm that this commercial affiliation does not alter your adherence to all PLOS ONE policies on sharing data and materials by including the following statement: ""This does not alter our adherence to PLOS ONE policies on sharing data and materials.” (as detailed online in our guide for authors http://journals.plos.org/plosone/s/competing-interests) . If this adherence statement is not accurate and there are restrictions on sharing of data and/or materials, please state these. Please note that we cannot proceed with consideration of your article until this information has been declared.

Reply

Thank you for the comment. Some of the authors, H.U., K.S., R.A., K.H., S.T., and M.I. are HOMER ION Co., Ltd. employees, however the authors believe there are no competing interests arising from this affiliation. This does not alter the authors' adherence to all the PLOS ONE policies on sharing data and materials. There are no planned patents or commercial products resulting from this work.

Funding and competing interests were added to the cover letter.

Comment 8

8. Please include your full ethics statement in the ‘Methods’ section of your manuscript file. In your statement, please include the full name of the IRB or ethics committee who approved or waived your study, as well as whether or not you obtained informed written or verbal consent. If consent was waived for your study, please include this information in your statement as well.

Reply

Thank you for the comment. The Ethics Review and approved No. for Animal Experimentation were added to the Animal section of Materials & Methods.

Comment 9

Reply

Thank you for the comment. The reviewers' comments did not recommend citing any specific published papers.

When additional citations were necessary for the manuscript, I cited papers of my own choosing as appropriate.

Reviewer #1 (Comments to the Author (Required)):

Comment 1

1. The introduction is redundant; it is unclear in L55-81 whether this study is an extension of your research or whether you are trying to establish a new experiment based on previous research. The aim of the study should be clearly indicated.

Reply

We greatly appreciate your valuable comments. We have revised the Introduction, manuscripts according to your comments.

Comment 2

2. A variety of factors can cause muscle atrophy, but the authors used the peripheral neuropathy model in this study. However, there is no research introduction on the use of this model. Please add this point to the Introduction section.

Reply

Thank you for the comment. A research introduction to the denervation model has been added to the Introduction.

Comment 3

3. The Methods and Materials section describes the animals as follows: "Acute stimulation experiment, chronic stimulation experiment, n=52." Could you please provide the specific number of animals in each group? Similarly, when stating that acute stimulation experiments were performed in three groups and chronic stimulation experiments were divided into four groups, the number of rats in each group should be specified.

Reply

Thank you for the comment. Added the specific number of animals in each group to Materials & Methods.

Comment 4

4. For Western blotting, is a loading control or total protein amount quantified at the same time to eliminate differences in protein expression between samples and the effects of manipulation during the experiment? The authors should add these to the Material & Methods section.

Reply

Thank you for the comment. In the Western blot, total protein was quantified using ponseau and normalized. I added this to the manuscript as per your comment.

Comment 5

5. The authors described “Stimulation intensity was preliminarily tested with a belt electrode and set to a minimum current run value of 3.0 mA (1.2 mA for twitch), which produced maximum torque with 60 Hz stimulation.” The authors should add the results of the preliminary investigation to Material & Methods section (or supplemental files).

Reply

We greatly appreciate your valuable comments. The results of a preliminary study on current strength are added to Figure 1b.

Comment 6

6. In “Acute response to electrical stimulation with belt electrodes” section, why did the authors examine the glycogen content and phosphorylated AMPK immediately after exercise and phosphorylated p70S6K in 6 h after exercise? Please provide a reason for the time to harvest the skeletal muscles. The authors did not assess the phosphorylated AMPK and p70S6K in TA. This investigation was titled “multiple muscle groups”, the lack of these results is a primary concern for this study.

Reply

We greatly appreciate your valuable comments. AMPK activation is observed during and immediately after exercise, and then rapidly returns to the basal level.( Am J Physiol 270: E299E304.) Phosphorylation of p70S6K (mTOR) gradually and significantly increased up to 6 hours after exercise cessation. Therefore, AMPK analysis was performed immediately after stimulation, and phosphorylation of p70S6K analysis was performed 6 hours later. We also performed the same experiments in TA muscles.

References were added to the manuscript.

Comment 7

7. After denervation, the sciatic nerve may sometimes reattach. To avoid this, the proximal and distal portions are often removed a few millimeters from the amputated area. Did the authors perform this procedure? Or, did they confirm that no reattachment occurred by dissection?

Reply

Thank you for the comment. The sciatic nerve was removed by a few millimeters. We have revised the Materials Methods manuscripts according to your comments.

Comment 8

8. Since the total RNA concentration differs from sample to sample, the amount of RNA contained in the RNA solution (5 μL) differs for each sample. Fortunately, the authors have determined the total RNA concentration by Nanodrop; please provide the total RNA concentration of each sample used for the 18S and 28S searches and show that there is no difference between groups.

Reply

Thank you for the comment. The RNA concentration was calculated as the amount of RNA per muscle weight and corrected for muscle weight. Each RNA concentration was added as supplementary information.

Comment 9

9. Regarding RT-PCR, how did the authors quantify Atrogin-1 and MuRF1 mRNA expression　? Did you use the internal control? If so, the primer sequences should be listed in Table 1.

16. In Figure 6, did the authors use the loading control (e.g., GAPDH)

Reply

Thank you for the comment. In this experiment,β-actin was used as an internal control. Concerning GAPDH, the glycolytic reaction is involved in carbohydrate metabolism in skeletal muscle research. Thus, it is difficult to use as a method for addressing muscle atrophy. Since the amount of ribosomes is reduced in atrophied muscles (Kotani, et al. J Appl Physiol.2022), 18S is not suitable for our experiment. These are the reasons why β-actin was selected in this study.

As per your comment, we added to Materials & Methods and added primer sequences to the Table.

Comment 10

10. It is difficult to confirm inter-individual variation in the mean±standard error. Please indicate the data as mean±standard deviation (SD). The authors should also add about the post-hoc tests. Furthermore, much data was listed as folds. Please re-describe all the data in mean±SD.

Reply

Thank you for the comment. Materials & Methods, Figures, and Figure legends have been revised. Post-hoc tests have also been added to Materials & Methods, and graphs have been revised to show raw data or divided values, except for Western blots, electrophoresis, and RT-PCR for chronic stimulation. Relative comparisons were performed for Western blots, electrophoresis, and RT-PCR for chronic stimulation.

Comment 11

11. In Figure 3, the band images of p-p70S6K and p-AMPK are too weak to confirm; please re-prepare the band images along with t-p70S6K and t-AMPK. Furthermore, two bands can be identified in p-p70S6K. Which of the two bands did you adopt?

Reply

Thank you very much for the comment. The band image in Figure 3 was re-prepared. The band of p-p70S6K was shown in cell signaling, and the upper band with high intensity was used.

Comment 12

12. In Figure 4, why did you separate the muscle wet weight into right and left sides? I suppose they should be combined. In addition to muscle wet weight, please add data on body weight and relative weight ratio.

Reply

Thank you for the comment. Data for muscle mass weight are presented separately to show that both left and right sides of the body had reduced muscle atrophy. Body weight, relative body weight ratio were added to Figure 4.

Comment 13

13. In Figure 5, which color indicates which muscle fiber type? Furthermore, since the scale bar is not listed, the sizes cannot be compared. Why is this data listed as a percentage while the muscle wet weight is listed as raw data? All data should be presented as raw data, not as ratios.

Reply

Thank you for the comment. The relationship between color and muscle fiber was added to Fig legends. Scale bars were added, and graphs of raw data were replaced.

Comment 14

14. Which muscles did the authors use for analyses of COXIV, 18S, 28S, Atrogin-1, and MuRF1? You must indicate all the data of the above parameters in TA and GAS.

Reply

Thank you for the comment. Data for COXIV, 18S, 28S, Atrogin-1, and MuRF1 were used in GAS, which showed a significant muscle

---

## [Decision Letter · Decision Letter 1]

1 Oct 2025

Mixed twitch and tetanus electrical stimulation via belt-electrode effectively attenuates denervation-induced muscle atrophy.

PONE-D-25-28377R1

Dear Dr. Uno,

We’re pleased to inform you that your manuscript has been judged scientifically suitable for publication and will be formally accepted for publication once it meets all outstanding technical requirements.

Kind regards,

Atsushi Asakura, Ph.D

Academic Editor

PLOS ONE

Additional Editor Comments (optional):

Reviewers' comments:

Reviewer's Responses to Questions

**Comments to the Author**

1. If the authors have adequately addressed your comments raised in a previous round of review and you feel that this manuscript is now acceptable for publication, you may indicate that here to bypass the “Comments to the Author” section, enter your conflict of interest statement in the “Confidential to Editor” section, and submit your "Accept" recommendation.

Reviewer #1: All comments have been addressed

Reviewer #2: All comments have been addressed

Reviewer #3: All comments have been addressed

2. Is the manuscript technically sound, and do the data support the conclusions?

Reviewer #1: Yes

Reviewer #2: Yes

Reviewer #3: Yes

3. Has the statistical analysis been performed appropriately and rigorously? 

Reviewer #1: Yes

Reviewer #2: Yes

Reviewer #3: Yes

4. Have the authors made all data underlying the findings in their manuscript fully available?

Reviewer #1: Yes

Reviewer #2: Yes

Reviewer #3: Yes

5. Is the manuscript presented in an intelligible fashion and written in standard English?

Reviewer #1: Yes

Reviewer #2: Yes

Reviewer #3: Yes

6. Review Comments to the Author

Reviewer #1: Based on my previous comments, I carefully reviewed the revised manuscript prepared by the authors. The authors have sincerely addressed my concerns, and improvements to the manuscript have been confirmed on all points. I believe this manuscript has achieved a quality worthy of publication in PLOS ONE. I hope this work will contribute to the development of new intervention strategies for denervation-induced muscle atrophy.

Reviewer #2: The manuscript “Mixed twitch and tetanus electrical stimulation via belt-electrode effectively attenuates denervation-induced muscle atrophy” presents a novel strategy where combined twitch and tetanus stimulation more effectively preserves skeletal muscle mass than tetanus alone. The study highlights that this approach enhances both mitochondrial biogenesis and protein synthesis, addressing key pathways in denervation-induced atrophy. Importantly, the authors carefully revised the manuscript to address all editorial and reviewer concerns, including clarifying methodology, improving figures, ensuring statistical rigor, and expanding discussion on mechanisms and clinical applications

Decision: Based on its novelty, translational potential, and thorough revisions, I recommend acceptance of this manuscript for publication in PLOS ONE.

Reviewer #3: Overall comment

The manuscript has been revised, and the logic is clear and easy to follow. With a few additional revisions, it could be made even clearer.

Specific comments

Line 79–83

The manuscript states that aerobic and resistance exercise have been extensively studied. However, the outcomes of these studies are not described. Please provide a brief summary of the main findings.

Line 262

In the context of RT-PCR, the use of the term “loading control” is inaccurate. A more appropriate description would be “internal control” or “reference gene.”

In addition, could you please clarify which method was used for quantification in the RT-PCR analysis (e.g., ΔCt, ΔΔCt, or standard curve method)?

Figure 5

From the figure legend, my understanding is that the black dots represent the mean cross-sectional area of muscle fibers for each rat. However, this is not certain. Please clarify whether these values indicate the mean, the median, or another definition. Simply stating “cross-sectional area (CSA) of TA and GAS” is not sufficiently clear.

Muscle wet weights

This manuscript presents relative muscle wet weights, suggesting that individual variability has been considered. However, I would appreciate it if you could also provide the individual body weights (g) and absolute muscle wet weights (mg) of each rat.

Western blot images

Is it correct to assume that the Western blot images correspond one-to-one between the upper and lower panels? If so, please indicate this clearly in the supplementary data.

Muscle fiber type composition ratio (supplementary data)

It is difficult to distinguish between type IIa and IIb due to the color similarity. In addition, the upper edge of the graph appears to be cut off. Please adjust the legend order to match the sequence displayed in the stacked bar chart. At present, the arrangement of Type I, IIa, and IIb in the legend does not correspond to their order in the chart.

Total muscle fiber number

In addition, is it correct that the total number of muscle fibers showed no statistically significant differences between the different electrical stimulation protocols?

7. PLOS authors have the option to publish the peer review history of their article (what does this mean? ). If published, this will include your full peer review and any attached files.

**Do you want your identity to be public for this peer review?** For information about this choice, including consent withdrawal, please see our Privacy Policy .

Reviewer #1: No

Reviewer #2: No

Reviewer #3: No

---

## [Editor Report · Acceptance letter]

PONE-D-25-28377R1

PLOS ONE

Dear Dr. Uno,

I'm pleased to inform you that your manuscript has been deemed suitable for publication in PLOS ONE. Congratulations! Your manuscript is now being handed over to our production team.

Kind regards,

on behalf of

Dr. Atsushi Asakura

Academic Editor

PLOS ONE